# Role of Excitatory Amino Acid Carrier 1 (*EAAC1*) in Neuronal Death and Neurogenesis After Ischemic Stroke

**DOI:** 10.3390/ijms21165676

**Published:** 2020-08-07

**Authors:** Minwoo Lee, Dong Gyun Ko, Dae Ki Hong, Man-Sup Lim, Bo Young Choi, Sang Won Suh

**Affiliations:** 1Department of Physiology, Hallym University, College of Medicine, Chuncheon 24252, Korea; minwoo.lee.md@gmail.com (M.L.); kdkhiah@gmail.com (D.G.K.); zxnm01220@gmail.com (D.K.H.); 2Department of Neurology, Hallym Neurological Institute, Hallym University Sacred Heart Hospital, Anyang 431070, Korea; 3Department of Medical Education, Hallym University, College of Medicine, Chuncheon 24252, Korea; ellemes@hallym.ac.kr

**Keywords:** excitatory amino acid carrier 1, ischemia, neuron death, neurogenesis, cysteine, glutathione, reactive oxygen species

## Abstract

Although there have been substantial advances in knowledge regarding the mechanisms of neuron death after stroke, effective therapeutic measures for stroke are still insufficient. Excitatory amino acid carrier 1 (*EAAC1*) is a type of neuronal glutamate transporter and considered to have an additional action involving the neuronal uptake of cysteine, which acts as a crucial substrate for glutathione synthesis. Previously, our lab demonstrated that genetic deletion of *EAAC1* leads to decreased neuronal glutathione synthesis, increased oxidative stress, and subsequent cognitive impairment. Therefore, we hypothesized that reduced neuronal transport of cysteine due to deletion of the *EAAC1* gene might exacerbate neuronal injury and impair adult neurogenesis in the hippocampus after transient cerebral ischemia. *EAAC1* gene deletion profoundly increased ischemia-induced neuronal death by decreasing the antioxidant capacity. In addition, genetic deletion of *EAAC1* also decreased the overall neurogenesis processes, such as cell proliferation, differentiation, and survival, after cerebral ischemia. These studies strongly support our hypothesis that *EAAC1* is crucial for the survival of newly generated neurons, as well as mature neurons, in both physiological and pathological conditions. Here, we present a comprehensive review of the role of *EAAC1* in neuronal death and neurogenesis induced by ischemic stroke, focusing on its potential cellular and molecular mechanisms.

## 1. Introduction

Stroke is one of the leading causes of death in the world and is a major cause of serious disability, accounting for 5.5 million deaths annually, with 44 million physical disabilities worldwide [1]. The relationship between oxidative stress and cell death has been an important issue under continuous study. The common cause of cellular oxidative stress includes excessive production of reactive oxygen species (ROS), which can lead to neuronal death [2]. Research on this process has revealed that one of the genes that promote antioxidation in humans is excitatory amino acid carrier 1 (*EAAC1*), and it carries out this function by increasing the cysteine concentration in the cell. *EAAC1*, also known as SLC1A1 or excitatory amino acid transporter 3 (*EAAT3*), increases intracellular glutathione (GSH) levels by importing cysteine from the extracellular space. The process includes the expression of the gene that affects neurons [3,4]. Thus, there is a prominent relationship between cell death in the central nervous system (CNS) and *EAAC1*.

Neurogenesis in the CNS of adult mammals occurs in the subventricular zone (SVZ) of the lateral ventricle, which is a part of the olfactory system, and the subgranular zone (SGZ) of the dentate gyrus in the hippocampus. The neuron-generating process in the former region includes neuronal stem cell (NSC) migration and integration in cells in the olfactory bulb, forming related interneurons and oligodendrocytes, including corpus callosum oligodendrocytes [5,6]; the latter process is closer to age-dependent memory/learning-related cell proliferation, forming dentate granule neurons and astrocytes. Radial glia-like NSCs transform into intermediate progenitor cells (IPCs) and then to neuroblasts, migrating to the granule cell layer to mature and become dentate granule neurons. It has been predicted that these NSCs and their sequences occur naturally to provide more plasticity to the brain, especially during learning processes [7].

Cerebral ischemia accounts for more than 70–80% of all strokes, originating from insufficient blood flow to deliver oxygen and nutrients to certain parts of the brain, unlike intracranial hemorrhage. Reperfusion after ischemia also results in further neuronal damage. A notable fact is that ROS are excessively produced in the reperfusion process, during which oxygenated blood rushes into cells adapted to hypoxic environments [8]. The aim of this review is to offer an overview of the current knowledge about the role of *EAAC1* in neuronal death and neurogenesis following ischemic stroke and the reperfusion process, and it explores how certain factors assist the gene’s functions. Focus is placed on the special contribution to neuron death and neurogenesis by the uptake of cysteine and GSH in aspects of their roles as antioxidants.

## 2. *EAAC1*: Its Relationship with Oxidation

### 2.1. The EAAC1-mediated Uptake of Glutamate and Cysteine

*EAAC1* is one of the five sodium-dependent *EAATs* that regulate glutamate transport in the CNS. Both glutamate/aspartate transporter (GLAST, also named *EAAT1*) and glutamate transporter 1 (GLT-1, also named *EAAT2*) are mainly expressed in glial cells (such as microglia, astrocyte, and oligodendrocytes), while GLT-1 is also found in various populations of neurons [9,10]. Together with *EAAT4* and *EAAT5*, *EAAC1* has been reported to be expressed in neurons from various regions [3]. Its concentration is about 100 times less than that of GLAST or GLT-1; however, it is heavily expressed in the hippocampus, with a concentration of 2- or 3-fold compared with that in other parts of the brain. We also found that EAAC1 is expressed in immature and mature neurons (Figure 1). Like other transporters, *EAAC1* is moved to the plasma membrane in a Rab11-dependent manner when needed, and SNAP-23 activates the transporter in the membrane. *EAAC1* modulates the activation of GluN2B-containing N-methyl-D-aspartate (NMDA) receptors to regulate the abundance of α-amino-3-hydroxy-5-methyl-4-isoxazolepropionic acid (AMPA) receptors in presynaptic/postsynaptic locations, which highlights its role in glutamate metabolism [11]. *EAAC1* is found to be more apparent in dendrites than in terminals, and its expression levels change as rats develop, which differs from two other glutamate transporters, GLAST and GLT-1 [11,12]. In addition, *EAAC1* exists in glutamatergic neurons as well as GABAergic neurons [11], where *EAAC1* may help maintain stable GABA levels by transporting glutamate for GABA synthesis. These phenomena suggest that EAAC1 is abundant in presynaptic/postsynaptic membranes [13].

However, immunolabeling experiments demonstrated that *EAAC1* is responsible for 1% of all glutamate uptake, whereas GLT-1 and GLAST are responsible for around 80% and 20%, respectively. The experimental results of the study showed that there was no change in the number of GLT-1 or GLAST when *EAAC1* was knocked out, proving that the role performed by *EAAC1* differs from that of other types of glutamate carriers [14]. Studies have shown that the sites of densely expressed *EAAC1* are the small intestine and the kidney, where it aids in the absorption of amino acids such as glutamate and aspartate [11]. In studies focused on its role in glutamate metabolism with knock-out models including *EAAC1^−/−^*, *GLAST^−/−^* and double mutant *GLAST^−/−^EAAC1^−/−^* show no significant differences in measured behaviors or neurodegeneration, so the carrier seems to play an important and perhaps unique role as an amino acid absorber [15,16].

In addition, from an experiment using *EAAC1*^−/−^ mice, it is clear that *EAAC1* is one of the major factors with oxidation resistance due to its cysteine transport function [14]. Critically, a comparative study between EAAT1, EAAT2, and EA−AT3 (*EAAC1*) and their L-cysteine uptake showed a dramatically selective aspect of *EAAC1* in cysteine transport. Tests with oocytes and anionic L-cysteine showed that, among EAATs, *EAAC1* is the most reactive. In contrast to intake through the same carrier with glutamate, the altered ionic state of cysteine had no effect on affinity or the transport rate through the carrier [17].

Therefore, *EAAC1* has two main functions, namely, the uptake of glutamate and cysteine, and the latter is the more predominant role that contributes to an intracellular GSH increase by pathways discussed later in this paper. Uptake of cysteine through *EAAC1* is considered to hasten the facilitation of glutamate transport by acting as a competitive inhibitor and lessening substrate affinity [18].

### 2.2. EAAC1 and Glutathione

Glutathione (GSH), a tripeptide that consists of glutamate, cysteine, and glycine, is essential for reducing oxidative stress in cells. It was first known for its anti-toxic reaction via GS-S-transferase, effluxing in forms of GSH-adducts. These two substances act as GSH-peroxidase and protect cells from toxic peroxides such as lipid hydroperoxides. The process results in glutathione disulfide (GSSG), and this replenishes cellular GSH through its reaction with the NADPH-dependent enzyme GSSG reductase. With some GSH-dependent enzymes, controlling dehydroascorbate reductase activities, maintaining the concentrations of ascorbic acid, and decreasing the free radical rate also show the relationship between GSH and its reductive features [19]. GSH also reduces superoxide, nitric oxide, hydroxyl radical, and peroxynitrite without the related enzymes. Its role in the formation of disulfides with protein thiol groups stops the irreversible oxidation of proteins as a form of S-glutathionylation [20].

GSH is synthesized in neurons as three constituent amino acids are imported. Two important ATP-consuming enzymes are required: GCL (glutamate-cysteine ligase, also known as γ-glutamylcysteine synthetase) catalyzes the rate-limiting step of γ-glutamylcysteine (γCysGly) formation, and GSH synthetase (GS) adds glycine to the dipeptide to complete the formation of GSH. Glutamine is imported through many EAATs, while cysteine is transported through *EAAC1*, and glycine levels are maintained by intracellular pathways [20]. Although three different amino acids are required, intracellular GSH is mainly regulated through the uptake of cystine (which transforms into cysteine in the cell) or other processes to uptake cysteine itself. The former process requires an antiport system of cystine and glutamate through system xc-, mostly in astrocytes, and is reported to regulate GSH concentrations in cerebrospinal fluid (CSF). Some immature neurons have been shown to use this mechanism to form GSH and were found to elevate its use during oxygenic conditions, so neurons seem to use this pathway when the cells must overcome oxidative pressure. Mature neurons import cysteine from glial cells in the form of GSH. Gap junction hemichannels are believed to be used during this transport, as GSH is cleaved into a γ-glutamyl moiety and a CysGly dipeptide through γ-glutamyl transpeptidase (γGT), and then into cysteine and glycine. The cysteine level is kept constant in neurons to produce GSH intracellularly. Thus, the production and use of glutathione in neurons must involve its interplay with astrocytes. To a lesser degree, neurons can uptake cysteine itself through EAATs called system XAG-. The neutral amino acid transporter system takes up ASC (alanine-, serine-, and cysteine-) through a sodium-dependent process. Neurons only have the system ASC transporter ASCT1, and glial cells have two of them [20,21].

As *EAAC1* promotes the intake of cysteine in neurons, and because cysteine is the limiting agent in the formation of GSH in neurons [3], it is evident that *EAAC1* is strongly related to oxidative stress in neurons via the production of GSH (Figure 2). The evidence of this relationship has been strengthened by experiments with *EAAC1* knock-out mice, which showed delayed onset of behavioral changes and gross atrophy compared to the wild type as a result of aging and their increased susceptibility to oxidants such as H_2_O_2_ and to 3-morpholinosydnonimine (SIN-1), with N-acetylcysteine (NAC) offsetting the damage. This result seems to arise from the NAC-promoted production of GSH [14]. Furthermore, experiments using ethanol in neurons, which impairs the expression of EAAC1 by 60–70%, revealed that the level of GSH and cysteine was reduced by up to 50% in primary cerebral cortical neurons. This effect was reversed by administration of NAC [22].

### 2.3. EAAC1 and Glutamate Transporter-Associated Protein 3-18 (GTRAP3-18)

Many factors are intertwined with the expression of *EAAC1*. Serum- and glucocorticoid-inducible kinase (SGK1) and phosphoinositide-dependent kinase (PDK1) promote *EAAC1* expression, and it is negatively controlled by the δ-opioid receptor, glutamate transporter-associated protein 3-18 (GTRAP3-18), and the phosphoinositide 3-kinase (PI3K) inhibitor wortmannin. Among them, GTRAP3-18 has been shown to be a major ER (endoplasmic reticulum) membrane protein for trafficking *EAAC1* [20,23], and this protein has shown active expression in many organs, including the brain [24].

GTRAP3-18 is also known as prenylated Rab acceptor (PRA) 1 family protein 3 (PRAF3) encoded from ADP Ribosylation Factor-Like GTPase 6 Interacting Protein 5 (ARL6IP5). By blocking and thus disabling serine 465 in the carboxy-terminal intracellular domain of *EAAC1*, this hydrophobic protein with four transmembrane domains lowers *EAAC1′*s affinity for glutamate. Previous studies have established this fact in in vitro models; in one such model, cerebellar granule neurons (CGNs) were treated with GTRAP3-18 antisense oligonucleotides cotransfected with green fluorescent protein (GFP). In vivo investigations in which intracellular GTRAP3-18 was modified supported this result. In addition, with its overriding effect over protein kinase C (PKC), GTRAP3-18 seems to be one of the dominant factors for regulating cellular GSH levels by lessening *EAAC1′*s affinity for cysteine. Tracking the activation of the proteins with sodium-dependent glutamate transport and GTRAP3-18′s nonactive interactions with other glutamate transporters also supports the result, proving that PKC controls the activation of *EAAC1* and that its deterioration by GTRAP3-18 causes a functional defect in *EAAC1* [18].

Although this might lead to increased cysteine import, the net result does not change. This is because GTRAP3-18 damages the cellular locating system of Rab1 [23]. Thus, *EAAC1* cannot be transported well to the cellular surface from the ER, and experimental studies have proved that the effects include reduced cellular GSH concentrations. Co-immunoprecipitation of c-Myc-tagged GTRAP3-18 with the carboxyl terminus of EACC1 but not with GLAST, GLT-1, or EAAT4 in in vivo models strengthens the evidence of this phenomenon [18]. Another study showed that with a low dose of methyl-β-cyclodextrin (MeβCD), which increases GTRAP3-18 expression but not to levels that induce toxicity, the level of GSH in cerebellar granule neurons (CGNs) dropped as the interaction between *EAAC1* and GTRAP3-18 increased, as determined by tracking GSH with ThioGlo-1 and the fluorescent dye CMFDA [21,25]. MeβCD administration is also known to decrease the level of EAAC1-mediated glutamate uptake by depletion of membrane cholesterol in primary cortical cultures [26].

## 3. Relationship Between *EAAC1* and Neuron Death Following Ischemic Stroke

### 3.1. Factors Related to Cell Death in the Reperfusion Process

As ischemia is induced in in vivo models, cellular homeostasis disruption occurs as a result of decreased pH and ATP, increased intracellular Na+ levels, and radical production. It leads to the formation of a “perpetrator,” which is a coined term that indicates the continuous damaging process of ischemia in the referenced paper. These include proteases, phospholipases, and free radical actions. These macromolecular changes lead to a critical malfunction that ends in cell death. According to the literature, most of these changes and damages are prominent in the CA1 and CA4 regions [27]. As cells adapt to the hypoxic state through the explained process, reperfusion leads to unregulated oxygenation conditions. Oxygen and hydrogen peroxide in blood reduce cellular iron or other transitional metals such as copper, inducing hydroxyl free radicals and subsequent tissue damage [8,28]. Increased free fatty acids, particularly arachidonic acid, from ischemia can also produce radicals as accumulated fats are metabolized by the reentered oxygen [29]. Many results in the literature support the definitive involvement of oxidative stress originating from the reperfusion state in neuronal death. Considered to be an earlier stage among perpetrators, intracellular protein oxidation from ischemia-derived inactivated glutamine synthetase was found to be selectively increased after ischemia [28]. Oxidative stress from ischemia–reperfusion can cause passive DNA damage, lipid peroxidation, mitochondrial dysfunction, and activation of apoptosis pathways. The reperfusion process especially activates ROS and reactive nitrogen species (RNS), worsening blood-brain barrier (BBB) breakdown and ending in cerebral edema or hemorrhagic transformation [30].

### 3.2. EAAC1 and Neuronal Death

Loss of oxidation-resistant proteins gives rise to neuronal death. As *EAAC1* is crucial for forming GSH, there is a strong positive association between *EAAC1* and neural death. Studies that show altered glutamine synthetase in the reperfusion process also indicate that the issue is related to GSH production [28]. In order to clarify the specifics of the relationship, an observation of diverse models without *EAAC1* and with neuronal death should be considered.

Some studies have detected rising L-cysteine levels but decreasing levels of glutathione in cerebral ischemia due to carotid artery occlusion in the hippocampus and striatum. Considering the acidic condition of the transporter due to the intracellular acidic environment caused by ischemia, researchers reported that a high cysteine level inside neurons thus prevents glutamate leakage, which can be caused by reverse transport, and neurons are able to escape glutamate toxicity, showing neuroprotective aspects [17,31]. In addition, *EAAC1*^−/−^ mice showed spatial learning and memory loss [16]. In studies that showed cysteine’s role as not only an antioxidant but also a metabolism promoter in cell signaling and ferroptosis, *EAAC1* was found to save neurons from ischemic disaster not only from antioxidants but also from deficient metabolism. Cysteine affects these through forms of cystine and GSH. Related cell signaling includes mammalian target of rapamycin complex 1 (mTORC1) signaling, activation of ISR (integrated stress response), and production of the neuroprotective gas H2S [32].

Our previous studies have demonstrated that genetic deletion of EAAC1 aggravates ischemia-induced neuronal death [33,34]. With 30 min of transient global ischemia in *EAAC1*^−/−^ mice, the number of degenerating neurons was increased in the cortex and hippocampus compared to the wild-type (WT) mice (Figure 3). However, NAC treatment significantly reduced the number of degenerating neurons in both WT and *EAAC1*^−/−^ mice. Currently used as a cure for acetaminophen overdose and thinning mucus in cystic fibrosis patients, NAC is a membrane-permeable cysteine precursor that does not need a unique transporter to be imported into neurons and also penetrates the BBB. Esterification of the carboxyl group of NAC leads to the production of N-acetylcysteine ethyl ester (NACET), and this lipophilicity-increased form of NAC was shown to be easily absorbed in rats [35]. These findings indicate that oxidative stress induced by ischemia is a crucial part of ischemia-induced damage and can thus be alleviated by increased antioxidant (in this case, glutathione) levels [33,34].

## 4. Recent Advances in the Relationship between *EAAC1* and Neurogenesis Following Ischemic Stroke

### 4.1. Neurogenesis: A General Understanding

The neurogenesis process is found primarily in two parts of the brain: One in the olfactory region of the subventricular zone (SVZ) and another in the subgranular zone (SGZ) of the hippocampus [36]. Their hyperexcitable traits and lower threshold for long-term potentiation induce embryo-originated neurons to stimulate their amplification of information coding in the human brain: NSCs in the dentate gyrus (DG) are related to long-term spatial memory and the discrimination of two similar patterns, and NSCs in the SVZ participate in short-term olfactory memory and flexible olfactory associative learning [37,38]. The increased number of granular cells in older brains and a blocked cellular division system do not lead to a decrease in the total number of granular cells, which suggests that neurogenesis is more than adjustments of the existing nervous system [36]. Considering the number of neurons in the human dentate gyrus (DG), almost all neurons are replaced throughout one’s life [39].

NSCs in these regions choose between quiescent and active states as they follow the cell cycle, and as they enter the latter stage, they undergo either (1) symmetric division resulting in two quiescent NCSs or two progenitors per cell or (2) asymmetric division through which they renew themselves, deciding which mature neuronal cell they will become. As adult SVZ NSCs (B1 cell) from radial glia grow and contact cerebrospinal fluid (CSF), they are relocated from the ventricular zone (VZ) to the SVZ by ependymal cells [40,41]. As they mature, migration to the olfactory bulb occurs through the rostral migratory stream, and differentiation into different subtypes of interneurons begins.

In the SGZ, glial fibrillary acidic protein (GFAP)-positive radial glial-like NSCs (type I cells) form intermediate progenitor cells (IPCs), which generate neuroblasts, adding granule cells by transforming while migrating to the inner granular cell layer in DG. The processes in this region are regulated by diverse regulators from ependymal cells or astrocytes or by blood circulating factors from other cells. One of the examples is sex-determining region Y-box 2 (Sox2), which is intensely activated in both regions of adult NSCs, or achaete-scute homolog 1 (Ascl1) [7,42]. GFAP-negative progenitors (Type II cells, also termed neuroblasts), which are found to form GABA synapses and regulate NCS growth, were also identified [43]. The cell will migrate to the stage of type III cells. Fully grown migrated neurons project their axons until reaching the CA3 region. It is known that Reelin signaling and DISC1 are pivotal factors for setting newborn granule cells [36,44].

The process of neurogenesis can be understood by tracking it with commonly used labelers for detecting particular cell stages. Immature newly born neurons can be detected by nucleotide analogs emitted during DNA replication, such as bromodeoxyuridine (BrdU) or [H3]-thymidine. These cells can also be marked by retrovirus-related substances that detect the M-phase of neurons, such as green fluorescent protein (GFP) or LacZ. Other substances, such as Ki67, collapsin response mediator protein 4 (CRMP4) (also known as TUC4), polysialic acid–neural cell adhesion molecule (PSA–NCAM), SOX gene families, and calretinin, are used to detect dividing neurons. Neurons that are differentiating can be detected by doublecortin (DCX), proopiomelanocortin (POMC), or neuronal nuclei (NeuN) antibodies. It has been reported that newly generated neurons in the dentate gyrus and striatum are GABAergic interneurons, with the former requiring two maturation steps and the latter needing only one [36,43,45,46]. The markers used to track down cellular mitosis in the rodent DG are as follows: Type I cells are marked by GFAP+Hes5+ or Nestin+Sox1+Sox2+BLBP (brain lipid-binding protein)+; Type II cells are marked by Nestine+Sox1+Sox2+BLBP+ or Mash1+ or Prox1+NeuroD+; type III cells are marked by NeuroD1+DCX+; and mature neurons are marked by NeuN+ [44].

### 4.2. Reperfusion State and Neurogenesis Regarding Oxidation

Although stem cells in the CNS are in comparatively low oxidative conditions compared to neurons in other organs [47], they are still exposed to a small number of oxygenic threats from cell metabolism. An increase in pathological markers, such as oxidized nucleic acid 8-hydroxyguanosine (8OHDG) and oxidized lipid 4-hydroxynonenal (HNE), and ROS production in neuropoietic sites illustrates that dividing cells are located in an endogenous oxidative stress environment, which is suspected of coming from the neurogenesis process itself [48]. Hence, the need for radical-fighting agents to protect them from becoming carcinogenic is deemed to be important [49]. This means that there exists a homeostatic and lowest possible level of oxygenic pressure for normal cell production, and an excessive amount of pressure will harm the neurons.

Because the reperfusion state causes the unregulated elevation of the oxidation level in low-pH and high-Ca^2+^ circumstances, it is clear that this imbalance impairs cellular division in the DG [50]. Inflicted ischemic damage is apparent after a certain period [27]. As a result of cerebral plasticity, the cell deaths from oxidative pressure sequentially lead to sudden increases in NSCs. However, to genuinely represent neurogenesis, those NSCs should develop into normal functioning neurons. Thus, behavior tests or molecular markers must be followed up after certain periods of time, usually 3–4 weeks [43,46,51].

Oxidative stress is an important factor for neurogenesis, as well as cellular GSH concentration. Nevertheless, a study showed that cystine transport by cystine/glutamate antiporter (xCT) is not essential for novel proliferation in the DG, which illuminates the importance of *EAAC1* as the main cysteine importer of cysteine in hippocampal regional mitosis. Our previous study showed that *EAAC1* is important for the survival of newborn neurons in the adult hippocampus under physiological and ischemic conditions [46]. We found that there are no differences in cell proliferation and neuronal differentiation in both young and aged *EAAC1*^−/−^ mice under ischemia-free conditions compared to the age-matched WT mice. However, *EAAC1*^−/−^ mice showed poor survival outcomes of newly generated neurons in both young and aged mice. In addition, *EAAC1*^−/−^ mice had fewer progenitor cells and their subsequent neuroblasts 7 days after ischemia. The neuronal maturation and survival rate of newly generated cells were also decreased in *EAAC1*^−/−^ mice 30 days after ischemia (Figure 4). This clearly shows the role of *EAAC1* in managing GSH levels in cells to control oxygenic stress.

In the sites where the pathological nucleic acid and lipids were analyzed, oxidative stress can be a problem in neurogenesis in the aspect of blocking proliferation of normal type I cells. Studies show decreased efficiency of pyruvate dehydrogenase and other types of mitochondrial dysfunction during the first 30–45 min of reperfusion, during which the number of hydroxyl radicals is at its maximum [50]; the membrane receptors or DNA repair enzymes can also be affected [52], so these clearly show that oxidative stress is affecting the ability of type I cells to complete the mitosis process. Past studies have shown an increased number of neurons during the reperfusion state but fewer active progenitor/neuroblasts, and these observations could indicate that the last part of the cell proliferation state is being inhibited. Additional study findings with BrdU+Ki-67+ progenitor cells further support this hypothesis as BrdU is found in the S-phase only but Ki-67 is expressed during all stages [53]. As an ischemic insult is inflicted, atypical migration happens, in which adult neurogenesis does not exist and thus fails to proliferate and integrate into damaged areas [43,54]. The migration and integration into a preexisting cell layer occur in type III cells.

## 5. Interventions that can Reinforce the Role of *EAAC1* After Stroke

### 5.1. Increasing the Expression or Effect of EAAC1

As *EAAC1* is one of the important neuroprotectants in the aspect of oxidative stress, manipulating factors related to *EAAC1* will multiply the effect of the gene on neural death and neurogenesis. The main interference involves factors that reinforce the expression of *EAAC1* in the cell membrane or lessen factors that impair the expression of *EAAC1*. As the expression of *EAAC1* is inhibited by GTRAP3-18, inhibition of GTRAP3-18 results in an increase in *EAAC1*. Many studies have revealed that protein expression can be induced by adding MeβCD, altering the cellular GSH level. This may be rooted in MeβCD elevating cAMP response element (CRE)-mediated gene expression and extracellular signal-related kinase (ERK) as cholesterol is cellularly discharged, both promoting the expression of GTRAP3-18 and physically impeding *EAAC1* [20,55]. As the uptake of cysteine and glutamate via *EAAC1* is done through the same pathway, finding that a cholesterol-rich glial culture medium increases glutamate transport means that the existence of extracellular cholesterol impedes the synthesis of GSH by changing the membrane composition of neural cells [56]. Some studies in the literature have stated that diminished levels of *EAAC1* are shown by the accumulation of metals such as Mn. A distinct negative correspondence between injected MnCl2 and mRNAs of the transporter, and the same variable with the amount of xCT-positive astrocytes, was detected in the striatum of the brain in neonatal mice. The phenomenon involves alteration of *EAAC1* and xCT directly by the metal ion itself, altering the cysteine level in the microcellular environment and increasing oxidative stress [57]. Therefore, substances that can chelate Mn ions would be helpful in promoting the effect of *EAAC1*.

### 5.2. Increasing the Level of GSH

Considering *EAAC1′*s role in producing GSH in neurons, the addition of a substance that can promote the cellular production of GSH itself will reduce cell death and increase functional neurons during the reperfusion state. Such an example is NAC. The effectiveness of NAC has been shown in many studies on the ischemia–reperfusion state to prove that defects of *EAAC1* are related to neural damage and neural production at the same time because NAC provides cellular cysteine [34,46]. The main impairment of oxidative stress in neurogenesis comes from depleting the cellular energy supply, and NAC can also intervene and elevate the proliferation rate.

Considering increases in mitotic factors such as basic fibroblast growth factor (bFGF) and inflammatory process inducers such as nitric oxide synthase, leukotrienes may promote a cellular deficit from ischemia [43]. In the aspect of oxidative stress, retinoic acid (RA) seems to also result in boosted GSH levels in neurons. A precursor substance of vitamin A, retinoic acid is mostly given in the form of the all-trans isomer of RA (ATRA). Metabolized by the RA receptor (RAR) and retinoid X receptor, RA was revealed to increase the activity of system Xc- and the cellular cystine level in neural stem cells in experimental mice given ATRA in vivo [58]. Given that an adequate amount of retinoic acid for balancing cellular oxidation is required for the normal proliferation and differentiation of neurons [59], a dose of retinoic acid after the ischemic-reperfusion state is predicted to benefit the compensative neurogenesis that controls cellular GSH levels.

## 6. Conclusions and Future Directions

Through deep insights about the relationship between *EAAC1* and both neuronal death and neurogenesis, it is evident that the oxidative stress from an outside stimulus is a critical factor in these phenomena. *EAAC1* was proved to have more significant functions in cysteine uptake for the synthesis of GSH, which, in turn, controls the oxidative level of cells. This was more evident in ischemia models. After an ischemic event, an increased level of GSH helped lessen the number of degenerating cells. The main mechanism is the diminishing of oxidative stress from the outside stimulus and saving neurons from the crisis. For neuronal plasticity, a controlled oxidative level increased the number of cells that correctly matured, and GSH seems to support the differentiation of new cells, therefore increasing the rate of real neurogenesis and the formation of functional cell layers.

Further studies should focus on whether a direct increase in *EAAC1* expression in neurons would bring about the same neuroprotective effects. Achieving this result would fortify current theories indicating a linkage between *EAAC1* and oxidative stress. For example, more studies should be performed on the relationship between GTRAP3-18 and cellular death or cellular regeneration. Understanding how it is linked to cytoskeletal changes during neuronal development gives insights into this aspect. Substances that are linked to fetal neural development, such as retinoic acids, need more investigation to uncover the role of oxidative stress in cellular processes that include not only correct maturation but also cellular development itself.

## Figures and Tables

**Figure 1 ijms-21-05676-f001:**
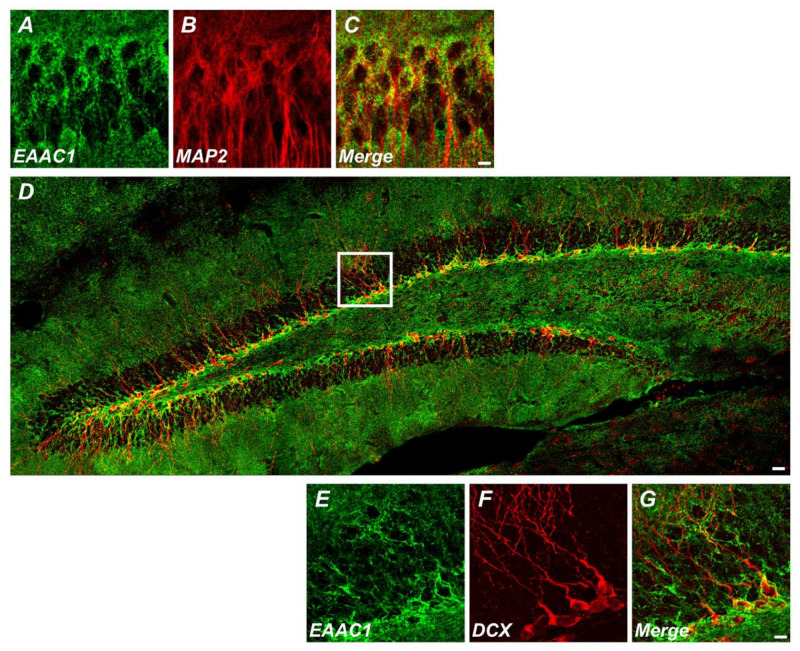
Excitatory amino acid carrier 1 (*EAAC1*) is expressed in immature neurons as well as mature neurons. (**A**–**C**) Immunofluorescence images representing the expression of *EAAC1*- (**A**) and MAP2-immunopositive cells (**B**) and *EAAC1*/MAP2 double-positive cells (**C**) in the hippocampal CA1 of wild-type (WT) mice (3–5 months old; weight 25–35 g). Scale bar = 5 µm. (**D**–**G**) Double-label confocal micrographs of *EAAC1* (green) and doublecortin (DCX) (red) in the hippocampal dentate gyrus (DG). Scale bar = 20 µm. (**E**–**G**) Higher magnification of the white box in **D** showing the expression of *EAAC1*- (**E**), DCX-immunopositive cells (**F**), and *EAAC1*/DCX double-positive cells (**G**) in the subgranular zone/glutamate-cysteine ligase (SGZ/GCL) of DG. Scale bar = 5 µm.

**Figure 2 ijms-21-05676-f002:**
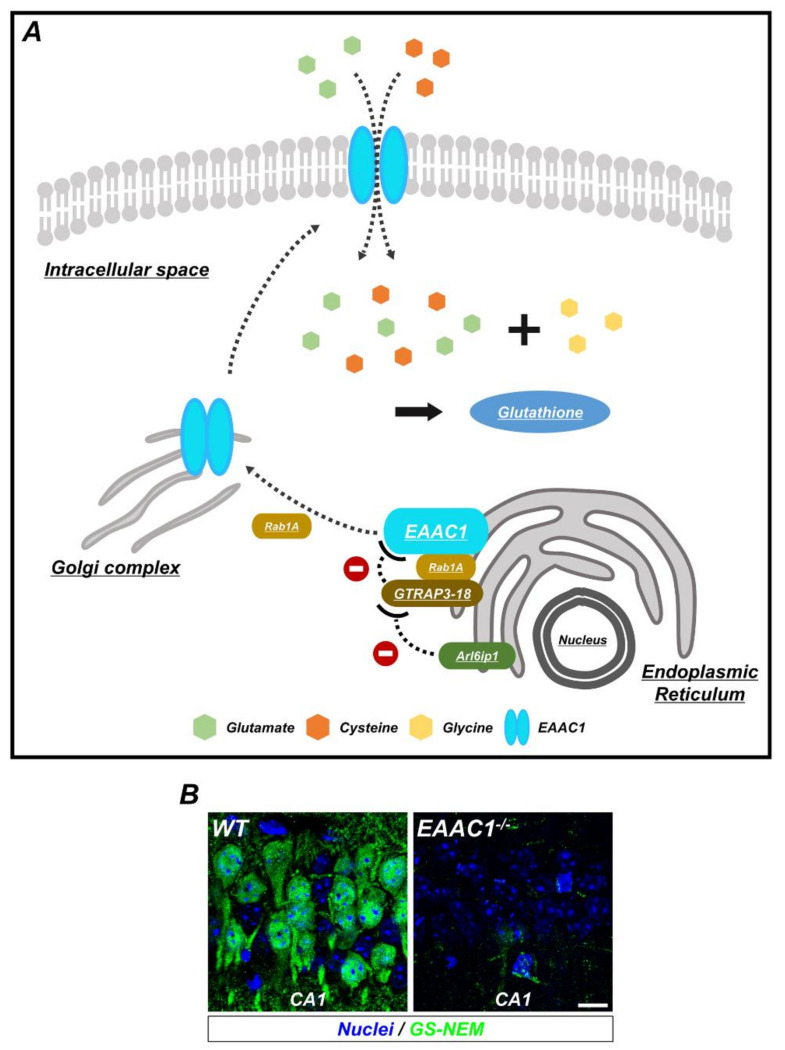
The synthesis of glutathione (GSH) by EAAC1/glutamate transporter-associated protein 3-18 (GTRAP3-18) in neurons. (**A**) GSH is composed of three amino acids: Glycine, cysteine, and glutamate. To increase the synthesis of GSH, EAAC1 is translocated to the plasma membrane and transports cysteine and glutamate into the neuron. Ras-related protein Rab1A regulates several membrane-trafficking pathways, including EAAC1 transport from the endoplasmic reticulum (ER) to the Golgi apparatus, which is interfered with when GTRAP3-18 binds to Rab1A. In addition, GTRAP3-18 directly retains EAAC1 in the ER to inhibit the synthesis of GSH. Thus, Arl6ip1, a GTRAP3-18-interacting protein, reduces GTRAP3-18/EAAC1 interaction and positively regulates EAAC1-facilitated glutamate transport. (**B**) Immunofluorescence images showing the GSH-N-ethylmaleimide (GS-NEM) adducts in the neurons of hippocampal CA1 from WT and *EAAC1*^−/−^ mice. *EAAC1*^−/−^ mice (3–5 months old; weight 25–35 g) showed neuronal GSH deficiency. Scale bar = 10 µm.

**Figure 3 ijms-21-05676-f003:**
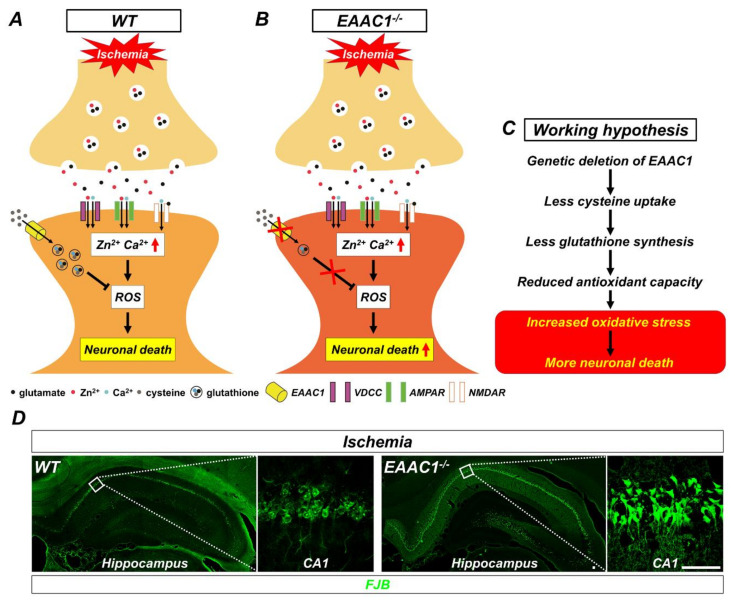
Proposed mechanism by which genetic deletion of EAAC1 increases ischemia-induced neuronal death. This illustration indicates several chain reactions that are thought to occur after the impairment of cysteine uptake into neurons as a result of EAAC1 gene deletion. (**A**) Glutamate and zinc are stored in presynaptic vesicles and released into synaptic clefts after ischemia. Excessive zinc can translocate into postsynaptic neurons and give rise to the disruption of calcium homeostasis, oxidative stress, and neuronal death. (**B**) Cysteine transport by EAAC1 is important for the synthesis of neuronal GSH. Reduced cysteine uptake due to EAAC1 gene deletion can lead to decreased GSH synthesis. (**C**) This can result in increased oxidative stress, which subsequently has a negative impact on the survival of neurons after ischemia. (**D**) Representative images showing Fluoro-Jade B (FJB)-positive cells, which indicate degenerating neurons, in the hippocampus of WT and EAAC1^−/−^ mice (3–5 months old; weight 25–35 g) at 3 days after ischemia (Insets show magnifications of white boxed areas). Scale bar = 50 µm.

**Figure 4 ijms-21-05676-f004:**
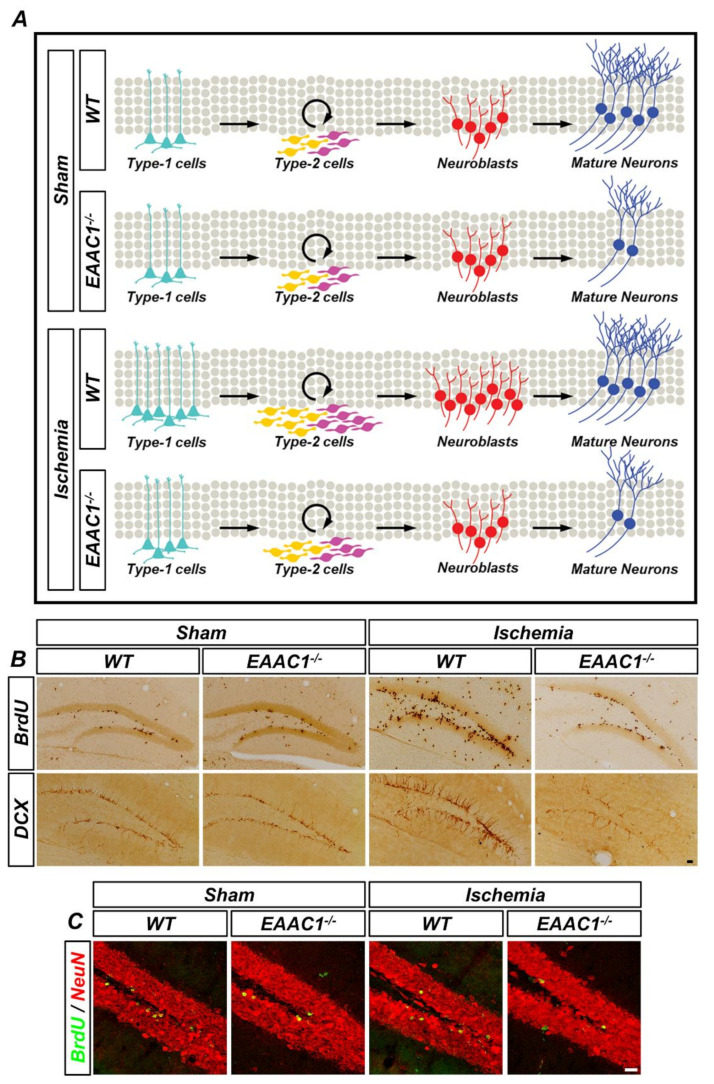
Genetic deletion of *EAAC1* reduces adult hippocampal neurogenesis after ischemia. (**A**) The schematic drawing shows that cell proliferation and neuronal differentiation are reduced in the *EAAC1*^−/−^ mice after ischemia, which means that EAAC1 gene deletion influences the entire process of adult neurogenesis. (**B**) Representative images showing the bromodeoxyuridine (BrdU)- and DCX-positive cells in the hippocampal DG from WT and EAAC1^−/−^ mice (3–5 months old; weight 25–35 g) at 7 days after ischemia or sham surgery. Scale bar = 50 µm. (**C**) Double-label confocal micrographs of BrdU (green) and neuronal nuclei (NeuN) (red) in the hippocampal DG from WT and EAAC1^−/−^ mice (3–5 months old; weight 25–35 g) at 30 days after ischemia or sham surgery. Scale bar = 20 µm.

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
