# Peer review of "Role of Excitatory Amino Acid Carrier 1 (*EAAC1*) in Neuronal Death and Neurogenesis After Ischemic Stroke"

_ijms, 2020, doi:10.3390/ijms21165676_

Round 1
Reviewer 1 Report
“Role of excitatory amino acid carrier 1 (EAAC1) in neuronal death and neurogenesis after ischemic stroke” by Lee et al is an interesting review of the role the neurotransmitter transporters role in stroke recovery. The authors have a unique view on the subject that could be better conveyed with more precise messaging – I’m still unclear if the authors support the idea that it is the function of EAAC1 as a cysteine carrier that is important, the genetic modulations that this transporter may stimulate (and a clearer description of how) or just a developmental profile that makes it an attractive therapeutic target for stroke-recovery?
Comments on the manuscript specifically:
Line 38 – “increasing glutamate concentration in the cell.” This is an inappropriate statement that is well discussed throughout the review that it’s cysteine – not glutamate.
Line 41 – EAAC1 is not typically found in astrocytes so the term “astroglial cells” is incorrect. The authors may wish to expand on the role of microglia and oligodendrocytes more specifically here rather than in the next paragraph.
Line 73: GLT1 is also found in populations of neurons as the work of Rosenburg’s lab and Danbolt’s have demonstrated that should be expanded on here.
Line 79: I do not think that the authors meant that it is the EAAC1 gene that modulates expressions, but rather the gene product or maybe the function (?). Otherwise, a greater explanation is needed here.
Line 82: The sentence ending with “an elevated number of glutamate receptors” is in need of a reference.
Line 84: the two other neurotransmitters should be named. I’m guessing that it’s GABA, since that is next discussed, but what is the other?
Line 89: Discussion in this paragraph regarding the knockout animals should include discussion of aged animals (Swanson lab) and the GLAST/EAAC1 combined knockouts too (Stoffel lab).
The primary data in Figures 1-4 should include the ages of the animals used. Figures 3D and 4B and C are lacking vehicle controls. These are particularly important in the observations regarding development/aging in the EAAC1 knockout animals as the Swanson studies (above) indicate
Figure 4 is particularly confusing – perhaps a timecourse as an x-axis would help to illustrate the time-course that the authors are discussing. The figure indicates that EAAC1 KO animals start with fewer Type-1 cells and therefore fewer mature neurons. This isn’t really due to the nature of the injury and subsequent neurogeneration and seems contrary to the authors thesis.
In the discussions of the effect of MeBCD the authors should look more carefully at other studies using cholesterol manipulation and the effects on neurotransmitter transporters. In particular, Butchbach et al, 2004 demonstrated that cholesterol modulates function of glutamate transporters (more so GLT1, but also EAAC1). DAT and GAT are also subject to altered function in response to changing membrane dynamics that has no immediate/direct effect on longer-term genetic changes.
Minor:
Line 184 – glutamate should be cysteine
The description of NAC on page 11 belongs earlier, on page 7 (line 238)
Line 360- cystine should be cysteine (cystine uptake is mediated by XCT, not EAAC1)
Similarly, line 397 also mistakenly refers to cystine.
Author Response
Comments on the manuscript specifically:
Line 38 – “increasing glutamate concentration in the cell.” This is an inappropriate statement that is well discussed throughout the review that it’s cysteine – not glutamate.
<Response: We appreciate this reviewer’s comments. We corrected this in the revised manuscript.>
Line 41 – EAAC1 is not typically found in astrocytes so the term “astroglial cells” is incorrect. The authors may wish to expand on the role of microglia and oligodendrocytes more specifically here rather than in the next paragraph.
<Response: We apologized there was a mistake. We corrected astroglial cells to neurons in the revised manuscript.>
Line 73: GLT1 is also found in populations of neurons as the work of Rosenburg’s lab and Danbolt’s have demonstrated that should be expanded on here.
<Response: We appreciate this reviewer’s comments. We added a new sentence and relevant references in the revised manuscript as follows: “Both GLAST (EAAT1) and GLT-1 (EAAT2) are mainly expressed in glial cells such as microglia, astrocyte, and oligodendrocytes), while GLT-1 is also found in various populations of neurons {Furness, 2008, 18805467}{Rimmele, 2016, 27129805}.”>
Line 79: I do not think that the authors meant that it is the EAAC1 gene that modulates expressions, but rather the gene product or maybe the function (?). Otherwise, a greater explanation is needed here.
<Response: We appreciate this reviewer’s comments. There has been a slight error in the process of English editing of the manuscript. We have revised the sentence to “EAAC1 modulates the activation of GluN2B-containing N-methyl-D-aspartate (NMDA) receptors to regulate the abundance of α-amino-3-hydroxy-5-methyl-4-isoxazolepropionic acid (AMPA) receptors in presynaptic/postsynaptic locations which highlights its role in glutamate metabolism.”
Line 82: The sentence ending with “an elevated number of glutamate receptors” is in need of a reference.
<Response: We appreciate this reviewer’s comments. We modified the sentence as stated above and added a relevant reference.>
Line 84: the two other neurotransmitters should be named. I’m guessing that it’s GABA, since that is next discussed, but what is the other?
<Response: We appreciate this reviewer’s comments and apologize if there was a mistake. In this sentence, the two other glutamate transporters are GLAST and GLT-1. Holmseth et al has studied the changes in expression of four different glutamate transporter subtypes, such as GLAST, GLT-1, EAAC1, and EAAT4, during brain development (E16 to adult). GLAST and GLT-1 were dramatically upregulated in the hippocampus with age, while there were only modest changes in the expression of EAAC1. They reported that EAAC1 levels increased slightly after birth and peaked at P14, before decreasing to approximately one-half over the next 6 weeks. Moreover, in order to link to the sentence that is next discussed, we corrected it in the revised manuscript as follows: “EAAC1 is found to be more apparent in dendrites than terminals, and its levels changed as rats developed, which are different from two other glutamate transporters such as GLAST and GLT-1 {Holmseth, 2012, 22539860;Aoyama, 2012, 23109897}. In addition, EAAC1 exists in both glutamatergic and GABAergic neurons {Holmseth, 2012, 22539860}.”>
Line 89: Discussion in this paragraph regarding the knockout animals should include discussion of aged animals (Swanson lab) and the GLAST/EAAC1 combined knockouts too (Stoffel lab).
<Response: We appreciate this reviewer’s comments. However, we have already discussed the study regarding age dependent neurodegeneration in the EAAC1-/- animals by Swanson lab in the next section 2.2 EAAC1 and glutathione, line 166-171. We have rephrased the paragraph and highlighted the changes. Upon your suggestions regarding GLAST-/-/EAAC1-/- models, we added a new phrases and relevant references in the revised manuscript as follows starting from line 97 : “In studies focused on its role in glutamate metabolism with knock-out models including EAAC1-/-, GLAST-/- and double mutant GLAST-/-EAAC1-/-, no significant differences in behaviors or neurodegeneration were observed, so the carrier seems to have a more important role as an amino acid absorber { Stoffel, 2004, 15363892;Peghini, 1997, 9233792}.”
The primary data in Figures 1-4 should include the ages of the animals used. Figures 3D and 4B and C are lacking vehicle controls. These are particularly important in the observations regarding development/aging in the EAAC1 knockout animals as the Swanson studies (above) indicate
<Response: We appreciate this reviewer’s comments. We added the ages of the animals used in figure legends of the revised manuscript. Figure 3D presented ischemia-induced neuronal death in both WT and EAAC1 knockout mice. This is Fluoro-Jade B (FJB) staining to detect the degenerating neurons. In the sham-operated groups, no FJB-positive cells were observed in the hippocampus. However, we added the representative images including sham-operated WT and EAAC1 knockout mice in Figure 4B and C of the revised manuscript.>
Figure 4 is particularly confusing – perhaps a timecourse as an x-axis would help to illustrate the time-course that the authors are discussing. The figure indicates that EAAC1 KO animals start with fewer Type-1 cells and therefore fewer mature neurons. This isn’t really due to the nature of the injury and subsequent neurogeneration and seems contrary to the authors thesis.
<Response: We appreciate this reviewer’s comments. We added an x-axis label in the figure 4 for a better description of the time-dependent steps of neurogenesis after ischemia. However, it indeed was our intention to state the vulnerability to oxidative stress in EAAC1-/- animals after ischemia. Sustained oxidative stress after ischemia results in less proliferation of type 1 cells and reduced survival rate during neuronal maturation.>
In the discussions of the effect of MeBCD the authors should look more carefully at other studies using cholesterol manipulation and the effects on neurotransmitter transporters. In particular, Butchbach et al, 2004 demonstrated that cholesterol modulates function of glutamate transporters (more so GLT1, but also EAAC1). DAT and GAT are also subject to altered function in response to changing membrane dynamics that has no immediate/direct effect on longer-term genetic changes.
<Response: We appreciate this reviewer’s comments. We added a new sentence and relevant references in the revised manuscript as follows: “MeβCD administration is also known to decrease the level of EAAC1-mediated glutamate uptake by depletion of membrane cholesterol in primary cortical cultures. {Butchbach, 2004, 15187084}.”
Minor:
Line 184 – glutamate should be cysteine
<Response: We corrected it in the revised manuscript.>
The description of NAC on page 11 belongs earlier, on page 7 (line 238)
<Response: We corrected it in the revised manuscript.>
Line 360- cystine should be cysteine (cystine uptake is mediated by XCT, not EAAC1)
Similarly, line 397 also mistakenly refers to cystine.
<Response: We corrected it in the revised manuscript.>
Reviewer 2 Report
This manuscript aims to provide a review of the excitatory amino acid carrier 1 (EAAC1) and its role in stroke-related neuronal death and neurogenesis. The paper is an extension of the authors' previously published directly relevant work (e.g. Choi et al 2014 Int J Mol Sci 15:19444-57).
1) While the topic is potentially interesting, this review does not go much beyond previously established points. The authors should emphasise more the original aspects of their working hypothesis and conclusions.
2) The Figures with presumably original data (since no source is referenced) are not discussed fully and their significance is not always clear.
3) In addition to stroke, there are several other conditions that can lead to excessive production of reactive oxygen species. Therefore it is not clear why the the role of EAAC1 is limited to stroke.
4) The different topics in the introduction are poorly linked.
5) Page 2, lane 79: The current IUPHAR-recommended nomenclature should be used for ionotropic glutamate receptors (e.g. GluN2B instead of NR2B).
6) Page 2, lanes 84-88: These sentences are unclear, more explanation is needed.
7) The text is often repetitive and there is little progression in the argument beyond re-stating several times that EAC1 promotes the intake of cysteine in neutrons and this will increase GSH formation.
8) The review focuses too much on the authors' previous studies, a broader and more inclusive approach would be preferable.
9) The link between neurogenesis and stroke is not explained clearly. The discussion ion this topic is often limited to technical issues (e.g. on page 9, lanes 286-302).
10) Use superscript for Ca2+
11) Conclusions are too general and not particularly novel or original.
Author Response
1) While the topic is potentially interesting, this review does not go much beyond previously established points. The authors should emphasise more the original aspects of their working hypothesis and conclusions.
<Response: We appreciate this reviewer’s comments. During revision with respect to this reviewer’s suggestions, we have emphasized the original aspects of our working hypothesis throughout the revised manuscript and modified the conclusions section as suggested.
2) The Figures with presumably original data (since no source is referenced) are not discussed fully and their significance is not always clear.
<Response: We appreciate this reviewer’s comments and understand this reviewer’s concern. We presented unpublished data in the Figures of the present study. As this reviewer’s point, we changed the representative images in Figure 4 of the revised manuscript.>
3) In addition to stroke, there are several other conditions that can lead to excessive production of reactive oxygen species. Therefore, it is not clear why the role of EAAC1 is limited to stroke.
<Response: We appreciate this reviewer’s comments and understand this reviewer’s concern. As this reviewer’s point, epilepsy, hypoglycemia, traumatic brain injury (TBI) as well as stroke can lead to excessive production of ROS. However, as our aim of this review was to review the specific anti-oxidative function of EAAC1 both in neuronal death and following neurogenesis, we focused our review on the ischemia models. Other disease models may have multiple etiologies of neuronal injury other than oxidative deprivation. For example, TBI model poses direct mechanical injury and secondary neuronal injury of broad etiologies, not confined to ROS overproduction.>
4) The different topics in the introduction are poorly linked.
<Response: We corrected it in the revised manuscript.>
5) Page 2, lane 79: The current IUPHAR-recommended nomenclature should be used for ionotropic glutamate receptors (e.g. GluN2B instead of NR2B).
<Response: We corrected it in the revised manuscript.>
6) Page 2, lanes 84-88: These sentences are unclear, more explanation is needed.
<Response: We appreciate this reviewer’s comments. We rephrased the aforementioned sentences for clear elaboration of our intent in the revised manuscript as follows: “In addition, EAAC1 exists in glutamatergic neurons as well as GABAergic neurons {Holmseth, 2012, 22539860} where EAAC1 may help retain GABA levels by transporting glutamate for GABA synthesis. These phenomena suggest that EAAC1 is abundant in presynaptic/postsynaptic membranes {Mathews, 2003, 12657662}.”>
7) The text is often repetitive and there is little progression in the argument beyond re-stating several times that EAAC1 promotes the intake of cysteine in neurons and this will increase GSH formation.
<Response: We appreciate this reviewer’s comments. We may have repeated stating the core mechanism of action of EAAC1 throughout the review. It was our intent to emphasize the argument we pose. However, upon your suggestion, we have deleted a few redundant statements in the revised manuscript.>
8) The review focuses too much on the authors' previous studies, a broader and more inclusive approach would be preferable.
<Response: We appreciate this reviewer’s comments. We have once more thoroughly reviewed the most recent studies regarding the function of EAAC1 by searching Pubmed for “EAAC1” or “EAAT3” in the title fields. We were able to update some of the relevant studies in the revised manuscript as follows : Line 172 “Furthermore, experiments using ethanol which impairs the expression of EAAC1 by 60-70% in neurons revealed that the level of GSH and cysteine was reduced up to 50% in primary cerebral cortical neurons which was also reversed by administration of NAC {Patel, 2017, 29206135}.”>
9) The link between neurogenesis and stroke is not explained clearly. The discussion on this topic is often limited to technical issues (e.g. on page 9, lanes 286-302).
<Response: We appreciate this reviewer’s comments. We have deleted a number of too detailed technical statements in the paragraph the reviewer has mentioned. (original manuscript line 287-289). For clear elaboration of stroke and following neurogenesis, we have revised the figure 4 appropriately and added more statements in the next section “4.2 Reperfusion state and neurogenesis regarding oxidation” as suggested.>
10) Use superscript for Ca2+
<Response: We corrected it in the revised manuscript.>
11) Conclusions are too general and not particularly novel or original.
<Response: We appreciate this reviewer’s comments. We have deleted rather too general statements from the conclusions section and added more insightful future directions as suggested in the revised manuscript as follows: “Further studies should focus on whether direct increase of EAAC1 expression in neurons bring about the similar neuroprotective effects. Finding these would fortify current linkage between EAAC1 and oxidative stress.”>
Round 2
Reviewer 2 Report
The authors provided appropriate responses to my comments and the revisions improved the manuscript.